# Iodide manipulation using zinc additives for efficient perovskite solar minimodules

Md Aslam Uddin ●[1], Prem Jyoti Singh Rana[1], Zhenyi Ni ●[1], Guang Yang ●[1], Mingze Li[1], Mengru Wang ●[1], Hangyu Gu ●[1], Hengkai Zhang[1], Benjia Dak Dou[2] & Jinsong Huang ●[1,3] ✉

Interstitial iodides are the most critical type of defects in perovskite solar cells that limits efficiency and stability. They can be generated during solution, film, and device processing, further accelerating degradation. Herein, we find that introducing a small amount of a zinc salt- zinc trifluoromethane sulfonate $(Zn(OOSCF_3)_2)$ in the perovskite solution can control the iodide defects in resultant perovskites ink and films. $CF_3SOO$ vigorously suppresses molecular iodine formation in the perovskites by reducing it to iodide. At the same time, zinc cations can precipitate excess iodide by forming a Zn-Amine complex so that the iodide interstitials in the resultant perovskite films can be suppressed. The perovskite films using these additives show improved photoluminescence quantum efficiency and reduce deep trap density, despite zinc cations reducing the perovskite grain size and iodide interstitials. The zinc additives facilitate the formation of more uniform perovskite films on large-area substrates ($78-108$ cm²) in the blade-coating process. Fabricated minimodules show power conversion efficiencies of 19.60% and 19.21% with aperture areas of 84 and 108 cm², respectively, as certified by National Renewable Energy Laboratory (NREL), the highest efficiency certified for minimodules of these sizes.

Despite the efficiency of small perovskite solar cells already reaching over 26%, the perovskite module efficiency is still far behind that of silicon modules[1–3]. To realize high-efficiency modules, it is essential to use scalable methods to fabricate efficient solar cells, and the resultant cell-to-module efficiency loss needs to be small. Minimizing the cell-to-minimodule efficiency loss needs to produce not only uniform perovskite films over a large area in ambient conditions but also good passivation uniformity and good uniformity of charge transport materials. Nonuniformity of the photoactive layers mainly impacts module short circuit $J_{SC}$ and fill factor ($FF$), despite the loss of averaged open circuit voltage ($V_{OC}$) of subcells from modules is sometimes observed.

Several efforts reported promising improvements of minimodules with small aperture areas by improving perovskite phase stability[4,5], hole transporting layer (HTL)/perovskite and perovskite/electron transporting layer (ETL) interfaces and their contact[6–8], and charge transport properties[7–9]. Even after all these efforts, relative cell-to-module efficiency loss is *ca.* 15-20% with an aperture area between $20-50$ cm²[6–10]. None of these studies addresses other important issues, such as iodide interstitial defects to reduce the cell-to-module efficiency loss. Nonuniformity arises from oxidized perovskite inks, especially oxidation of iodide to molecular iodine when inks are exposed to an ambient environment during the fabrication process in the ambient conditions. In addition, the introduction of 2D-iodide salts such as phenethylammonium iodide (PEAI) and dodecyl ammonium iodide (DDAI) in the perovskite inks increases more iodide and thus introduces more iodide interstitials. Both molecular iodine generation in inks and many iodide interstitials in the perovskite films present nonuniformity in perovskite films and thus lead to larger cell-to-module efficiency loss apart from the dead area and ITO resistive losses in minimodule design and fabrication[11].

Our study addresses these two issues by introducing zinc salts containing organic anions in the perovskite inks. In the zinc salt,

[1]Department of Applied Physical Sciences, University of North Carolina at Chapel Hill, Chapel Hill, NC 27599, USA. [2]CubicPV Inc., Bedford, MA 01730, USA. [3]Department of Chemistry, University of North Carolina at Chapel Hill, Chapel Hill, NC 27599, USA. ✉e-mail: jhuang@unc.edu

organic anions ($CF_3SOO^-$ ions) vigorously reduce the molecular iodine into iodides, whereas zinc cations can take away excess iodide by forming a Zn-Amine complex in the perovskite inks. Therefore, we are able to demonstrate large aperture area modules (78-108 cm²) with high reproducibility by improving the perovskite film quality and uniformity. Minimodules with an aperture area of 78-108 cm² show an average aperture area efficiency between 19.21% and 19.55%, with NREL-certified efficiency of 19.60% at a much larger aperture area of 79.67 cm², representing the highest efficiency of this size[12,13].

## Results and discussion
### Zinc salt additive and device performance
Metal salts and their complexes have been frequently used to tune the composition or modify the surface of perovskites to enhance solar cell performance, especially $V_{OC}$ and $FF$[14-18]. Previously, zinc halide salts ($ZnX_2$, X = Cl, Br, or I) were reported to improve the small device performance[16]. In this work, we used the zinc salts with different organic anions, including $-COO^-$, $-SOO^-$, and $-SO_2O^-$ anions with a strong affinity to the $Pb^{2+}$ ion[19]. We thus added different zinc salts, including zinc formate [$Zn(OOCH)_2$], zinc acetate [$Zn(OOCCH_3)_2$], zinc trifluoroacetate [$Zn(OOCCF_3)_2$], zinc trifluoromethane sulfinate [$Zn(OOSCF_3)_2$], and zinc trifluoromethane sulfonate [$Zn(OO_2SCF_3)_2$] as listed in Fig. 1a into the perovskite ink and evaluated their impacts on the device performances. To verify the effects of zinc compounds, we first fabricated small area devices with a mixed cation composition of methylammonium (30%)-formamidinium (70%) lead iodide ($FA_{0.3}MA_{0.7}PbI_3$) as an active layer with varying the concentration of each zinc salt by following our established blading procedure[20,21].

Figure 1b summarizes the performance statistics of the devices with optimized concentrations (Fig. S1 for other concentrations) of different zinc salt additives. The control devices deliver an average efficiency of 22.22 ± 0.69%, agreeing with the previous study[6,20]. Devices with zinc salt additives of $Zn(OOCH)_2$, $Zn(OOCCH_3)_2$, $Zn(OOCCF_3)_2$, $Zn(OOSCF_3)_2$, and $Zn(OO_2SCF_3)_2$ deliver the average

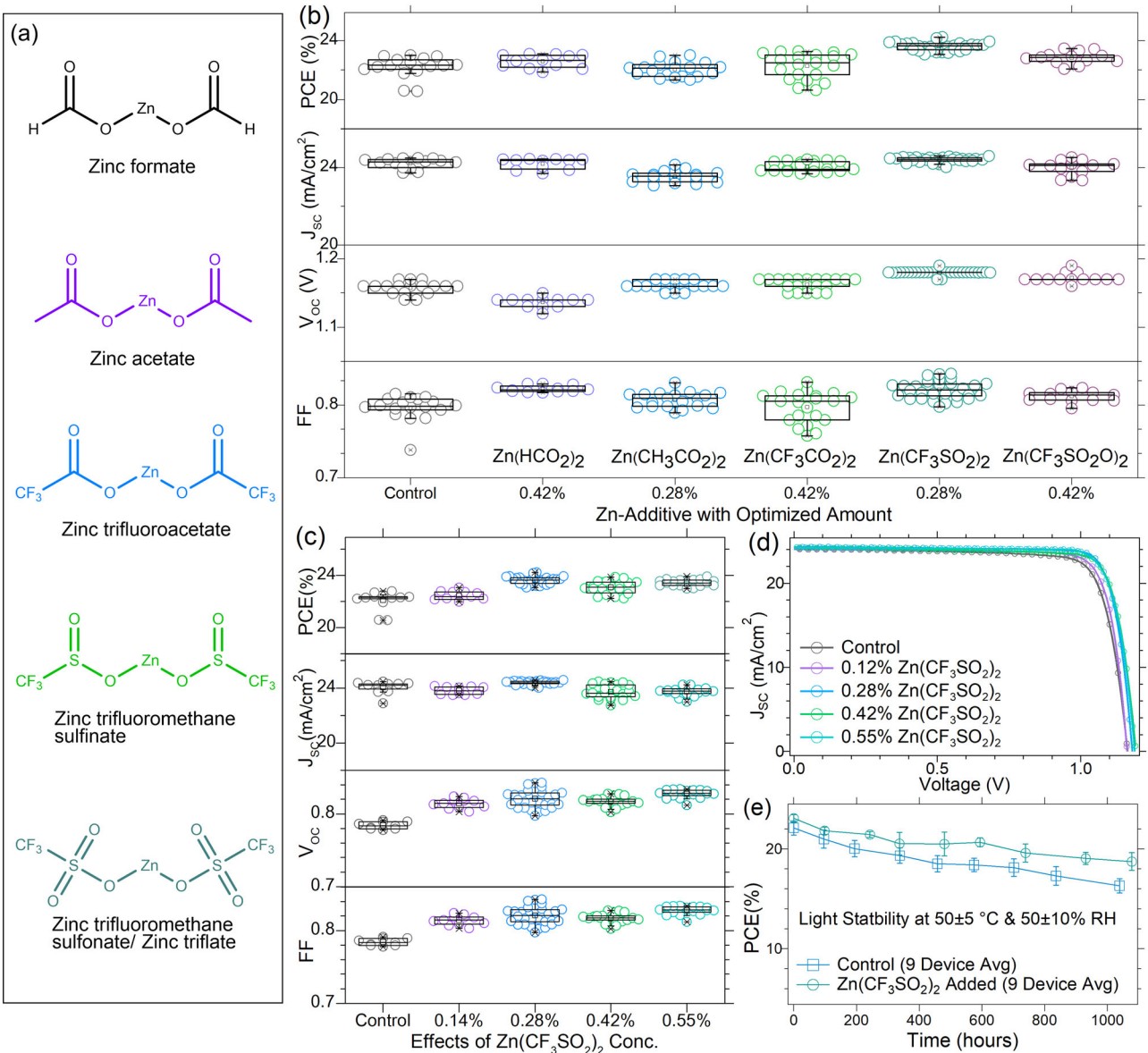

**Fig. 1 | Photovoltaic performance of small area cells. a** Molecular structures of zinc compounds as additives, **b** the performance statistics of the small devices with optimized concentrations of different zinc salt additives, **c** solar cell performance for small devices with varied $Zn(OOSCF_3)_2$ concentrations, **d** *J-V* curves small devices with varied $Zn(OOSCF_3)_2$ concentrations, and **e** time-dependent PCE of control and small devices with 0.42% $Zn(OOSCF_3)_2$ at $V_{OC}$ condition under one sun of LED light at 50 ± 5 °C and 50 ± 10% RH. There were nine devices tested for each type of device. The devices have an active area of 0.08 cm².

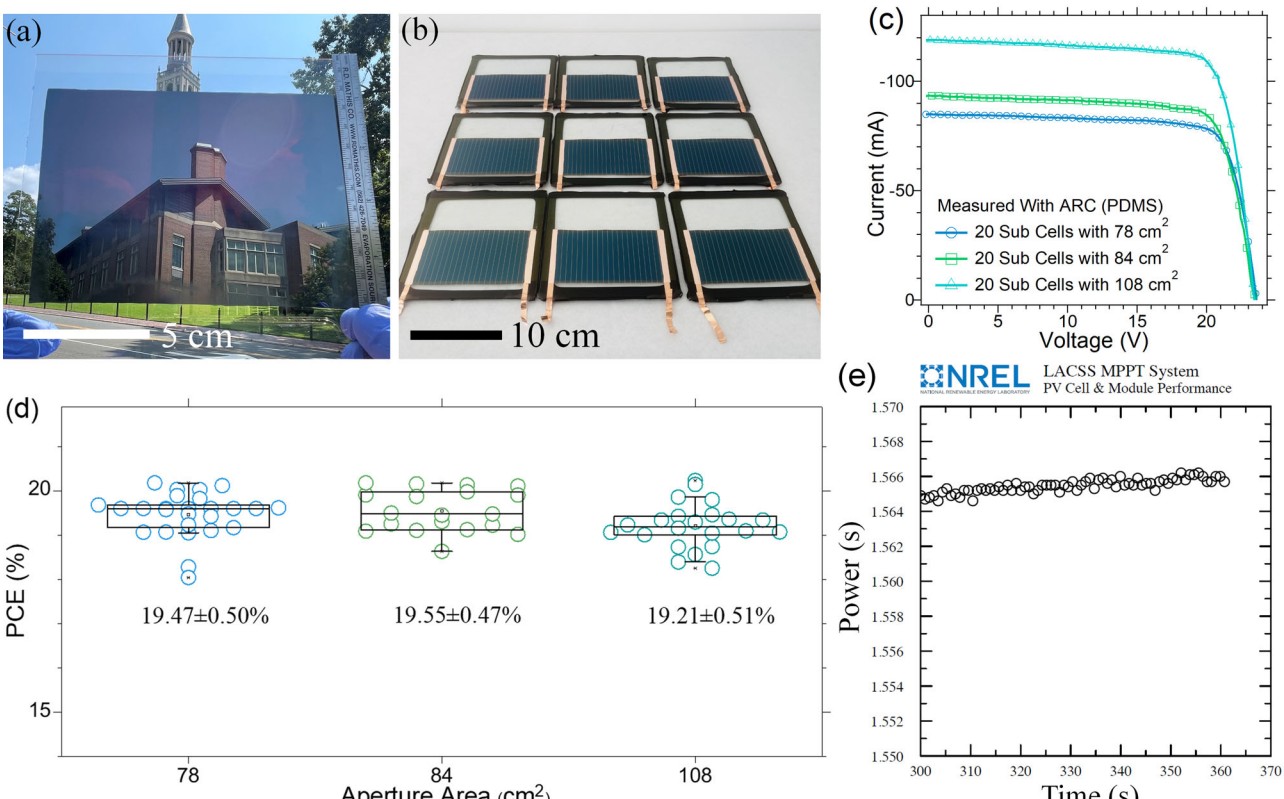

**Fig. 2 | Perovskite minimodule performance. a** A photograph of a bladed $FA_{0.3}MA_{0.7}PbI_3$ perovskite film on an ITO substrate with an area of ~130 cm², **b** a photograph of encapsulated minimodules with aperture areas of ~78 cm², **c** J-V curves of the minimodule with aperture areas of 78, 84, and 108 cm², respectively; **d** performance statistics of the minimodules with an aperture area of 78, 84, and 108 cm², respectively; and **e** the stabilized power output of the champion minimodules fixed at a bias of maximum power point for 360 s tested by NREL. Spectrum: ASTM G173 global, device temperature = 25.5 ± 2.0 °C, device area = 79.67 cm² ± 3.0%, and irradiance = 1000.0 W/m². The data is provided by NREL.

efficiency of 22.59 ± 0.44%, 22.03 ± 0.48%, 22.28 ± 0.86%, 23.61 ± 0.30%, and 22.83 ± 0.40% at specific molar concentrations (compared to the Pb-concentration) of 0.42%, 0.28%, 0.42%, 0.28%, and 0.42%, respectively. Among all the additives, $Zn(OOSCF_3)_2$ delivers the highest efficiency at almost all the additive concentrations. At the optimized concentration of 0.28% for $Zn(OOSCF_3)_2$, small devices are highly reproducible with all-around best device parameters, as shown by the *J-V* characteristics from seven devices shown in Fig. S2. All the device performance parameters from 30 devices from different batches are summarized in Table S1, which again confirms the excellent reproducibility of these high-performance small-area devices.

We studied the impact of $Zn(OOSCF_3)_2$ concentration on solar cell performance by analyzing how the photovoltaic parameters change. Figure 1c, d shows performance statistics and *J-V* curves of the solar cells with different concentrations of $Zn(OOSCF_3)_2$. With increased $Zn(OOSCF_3)_2$ concentration, the average $V_{OC}$ increased from 1.16 ± 0.01 V for the control devices to 1.18 ± 0.00 V for the devices with 0.28% $Zn(OOSCF_3)_2$, and the average FF increased from 0.80 ± 0.02 for the control device to 0.82 ± 0.01 for the devices with 0.28% $Zn(OOSCF_3)_2$. There was no clear trend for the variation of $J_{SC}$ with increased $Zn(OOSCF_3)_2$ concentration, despite the devices with optimal 0.28% $Zn(OOSCF_3)_2$ also improved slightly compared to the control devices. The results indicate that $Zn(OOSCF_3)_2$ improves the device efficiency through defect passivation as both $V_{OC}$ and $FF$ are significantly enhanced[22–25]. At the same time, slight increase in $Jsc$ is observed from a comparison of *J-V* characteristics and external quantum efficiency (EQE) spectra and integrated $J_{sc}$ of a control device and a target small device with 0.28% $Zn(OOSCF_3)_2$ added perovskites, as shown in Fig. S3.

We also tested the light-soaking stability of the small control devices and devices with $Zn(OOSCF_3)_2$ under one sunlight at open circuit conditions, following ISOS-L-1. The devices were encapsulated using epoxy and tested in the air with 50 ± 20% relative humidity, where the LED solar simulator light heated the devices to 55 ± 5 °C. The evolution of the efficiency with statistics is shown in Fig. 1e and Fig. S4. The devices with $Zn(OOSCF_3)_2$ retained 81.2% of their initial efficiency after light soaking for 1078 hours, while the control devices retained 72.8% of their initial efficiency after 1039 hours of light soaking, showing that zinc salt additives improved the light stability of perovskite solar cells.

After verifying the small device performance with the optimized concentration of $Zn(OOSCF_3)_2$, we blade-coated large area (>78 cm2) perovskite films and fand fabricated minimodules with an aperture area of 78-108 cm². The resultant perovskite films appear smooth and uniform (Fig. 2a and Fig. S5). Minimodules have the same structure and fabrication procedures as previously reported[20,21], and a photograph of several fabricated minimodules is shown in Fig. 2b. *J-V* characteristic curves of the minimodules with different aperture areas are shown in Fig. 2c. Minimodules with aperture areas of 78 cm², 84 cm², and 108 cm² delivered champion efficiencies of 20.18% ($V_{OC}$ = 1.17 V; $J_{SC}$ = 21.45 mA·cm⁻²; and $FF$ = 0.801), 20.18% ($V_{OC}$ = 1.18 V; $J_{SC}$ = 21.99 mA·cm⁻²; $FF$ = 0.796), and 20.23% ($V_{OC}$ = 1.19 V; $J_{SC}$ = 22.04 mA·cm⁻²; and $FF$ = 0.772), respectively. All the minimodules fabricated with 0.14-0.28% Zn(OOSCF3)2 also showed very reproducible performance among 67 fabricated minimodules. Average efficiencies of minimodules with aperture areas of 78, 84, and 108 cm² are 19.47 ± 0.50%, 19.55 ± 0.47%, and 19.21 ± 0.51%, respectively, as shown in Fig. 2d. The detailed photovoltaic parameters of all the

minimodules with different aperture areas are summarized in Table S2. Similar minimodule performance, regardless of the aperture area variation, indicates uniform and high-quality perovskite films facilitated by introducing $Zn(OOSCF_3)_2$- zinc compound. To verify their efficiency, we sent several minimodules to NREL for certification, and they showed a stabilized efficiency of 19.60% with an aperture area of ~80 cm², as shown in Fig. 2e.

This demonstration of larger area minimodules (80-110 cm²) with a stabilized aperture efficiency of 19.60% (active area efficiency of 20.66%) and champion efficiency of 20.20% (active area efficiency of 21.24%) is the record stabilized efficiency on ~100 cm² larger aperture area as summarized in the following Table S3. We have also tested the shelf stability of the same module certified by NREL. The module retains over 91% of its initial efficiency after storage for over 6 months. We then put the same module under 1 sunlight for 24 hours, and the module retains nearly 90% of its initial efficiency. The results of photocurrent-voltage (J-V) scans are shown in Fig. S6.

## Optoelectronic and morphologic property change by $Zn(OOSCF_3)_2$

To understand how $Zn(OOSCF_3)_2$ enhances the perovskite solar cell efficiency, we investigated the quality of $FA_{0.3}MA_{0.7}PbI_3$ perovskite thin films bladed on ITO/PTAA and carried out the optical and morphological characterization as shown in Fig. 3. As shown in Fig. 3a, among all the Zn additives, $Zn(OOSCF_3)_2$ improves the photoluminescence the most: it showed approximately 1.6 folds higher stead-state photoluminescence (PL) intensity than that of the control samples, indicating that Zn-additives reduce non-radiative recombination processes. PL quantum yield ($\Phi_{PL}$) and PL lifetime measurements were conducted on the control, and the perovskite films with $Zn(OOSCF_3)_2$ with a structure of ITO/PTAA/Perovskite, and $\Phi_{PL}$ of the film with $Zn(OOSCF_3)_2$ is 1.4 folds of the control film (0.46% vs. 0.66%). $FA_{0.3}MA_{0.7}PbI_3$ films with $Zn(OOSCF_3)_2$ also showed ~6 nm blue shift of PL peak as shown in Fig. 3b, indicating that $Zn(OOSCF_3)_2$ additives reduced the band-tail states[26,27]. The perovskite films with $Zn(OOSCF_3)_2$ also showed almost 3-times longer PL lifetime (0.7 μs for control film vs. 2.0 μs for 0.28% $Zn(OOSCF_3)_2$ added film), as shown in Fig. 3c. These studies conclude that $Zn(OOSCF_3)_2$ can passivate defects in the polycrystalline perovskites.

We further collected electroluminescence (EL) spectra of the control device and the devices with $Zn(OOSCF_3)_2$ added in the perovskites at an injection current of 24 mA/cm² (close to the $Jsc$ of the device measured at one sun illumination). As shown in Fig. 3d, EL intensity for the devices with $Zn(OOSCF_3)_2$ added films are ca. 2.5-fold higher than that of the control devices, indicating low non-radiative recombination losses that lead to the enhancement of $V_{OC}$. The estimated $V_{OC}$ increase from the enhancement of EL intensity compared to the control device is ~23.8 mV. The increase in $FF$ can be explained by the change in the series resistance ($R_S$) and larger shunt resistance ($R_{Sh}$) of the devices. As shown in Fig. S7 and Table S4, the champion devices with 0.28% $Zn(OOSCF_3)_2$ added perovskite films showed ca.1.4-fold higher $R_{Sh}$ and ca.1.1-fold lower $R_S$.

We also checked the morphology of $MA_{0.7}FA_{0.3}PbI_3$ perovskite films without/with $Zn(OOSCF_3)_2$ under scanning electron microscopy (SEM) to find out further whether $Zn(OOSCF_3)_2$ additive change grain growth behaviors. Figure 4a, b shows that the surface morphology of the perovskite films with $Zn(OOSCF_3)_2$ is much different from that of the control thin films. In addition, slightly smaller grain sizes were observed from the SEM cross-section images shown in Fig. 4c, d, suggesting small metal ions such as $Zn^{2+}$ likely increase the nucleation rate during the crystallization. Similar behaviors were observed for other Zn-additives as shown in Fig. S8. The X-ray diffraction pattern in Fig. 4e, f shows that the crystallinity of the film with $Zn(OOSCF_3)_2$ is not changed. Generally, a smaller grain size would likely create more grain

boundaries and thus induce more non-radiative charge recombination, which contrasts with the observed PL and EL enhancement and device performance improvement. This indicates the cations and anions have additional functions to passivate the introduced defects.

## The function of cations and anions of $Zn(OOSCF_3)_2$

We fabricated small devices with zinc halides as additives to verify if only $Zn^{2+}$ ions, organic anions, or both contribute to performance enhancement. As shown in Fig. S9, devices with $ZnCl_2$, $ZnBr_2$, and $ZnI_2$ showed comparable average efficiencies of $22.61 \pm 0.39\%$, $22.10 \pm 0.36\%$, and $22.45 \pm 0.40\%$ after optimization, respectively. These efficiencies of the devices with $ZnX_2$ additives are similar to that of the control devices but much lower than those with $Zn(OOSCF_3)_2$ as an additive, indicating that the organic anions played a critical role in enhancing the device efficiency.

It is known that $CF_3SOO^-$ ions can generate $SO_2$ and $CF_3^-$ in the presence of heat, acids, or bases[25], where $SO_2$ is a good reducing agent for iodine and triiodide reduction[28,29]. Gibbs free energy of the redox process between $SO_2$ and $I_2$ is ca. $-71$ k*J*mol⁻¹, indicating that the redox process between $SO_2$ and $I_2$ is spontaneous. Previous studies have shown that molecular iodine generation in perovskites under illumination is a common phenomenon and can accelerate the degradation of perovskites under operating conditions[20,30–32]. We thus hypothesize that $CF_3SOO^-$ ions can reduce $I_2$ into $I_3^-$ in the perovskite solutions[20]. To verify it, we added $Zn(OOSCF_3)_2$ into the iodine solution with a molar ratio of 1:1. We observed that the iodine color started to shift from dark brown to transparent (Fig. 5a). When the solution mixture of $I_2$ and $Zn(OOSCF_3)_2$ was put under the constant heat at 120 °C for over 15 minutes in the presence of water, or a mixture MAI and FAI (7:3), iodine color disappears, as supported by the absorption spectra in Fig. 5b. Therefore, the added $CF_3SOO^-$ ions can suppress $I_2$ formation in the perovskite solution, which is critical in the fabrication of perovskite solar cells in open air environment.

Since iodide interstitials form deep trapping states[33], and its concentration goes up with more iodide in the precursor solution, we examined the iodide-related deep trap density by the drive-level capacitance profiling (DLCP) and trap density of states ($tDOS$) by thermal admittance spectroscopy (TAS). The TAS and DLCP measurements can readily identify the presence of $I_i^-$ in the perovskites in the devices[34]. As shown in Fig. 5c, the $tDOS$ results reveal the trap density related to iodide interstitials slightly reduced in the $FA_{0.3}MA_{0.7}PbI_3$ polycrystalline thin film solar cells with the $Zn(OOSCF_3)_2$[35]. DCLP study reveals the deep trap density was reduced through the whole perovskite films as shown in Fig. 5d. The reduction of the deep traps in the perovskite thin films explains the passivation effect and the resultant increase in $V_{OC}$[35]. However, it cannot be explained by $CF_3SOO^-$ ions because $CF_3SOO^-$ ions cannot decrease the iodide interstitial concentration in the precursor concentration.

We then examined the function of zinc cations. It is reported that $Zn^{2+}$ can form the Z amines or primary ammonium salts[36–38]. Zn-amine complex can easily be observed as it precipitates out for a while in the perovskite precursor solution when $Zn(OOSCF_3)_2$ was added in the perovskite precursor solution in an excess amount (e.g., 2.5 mol% to Pb). Figure 5e shows the difference between the control perovskite precursor and precursor added with 2.5 mol% $Zn(OOSCF_3)_2$, where the white precipitate is the Zn-amine complex. In the Zn-amine complex, FA and MA are basic ligands confirmed by the Fourier transform infrared (FTIR) measurement as shown in Fig. S10. $Zn^{2+}$ is a strong Lewis acid that can deprotonate $FA^+$ and $MA^+$. Fig. S9a shows that the intensity of a broad peak around 1716 cm⁻¹ ($FA^+$) reduced dramatically after $Zn(OOSCF_3)_2$ addition, indicating that $Zn^{2+}$ deprotonates $FA^+$. Similarly, $Zn^{2+}$ also deprotonates $MA^+$, as indicated by the disappearance of the broad peak at around 1600 cm⁻¹ due to $MA^+$. To further verify if the Zn-amine complex can be formed from

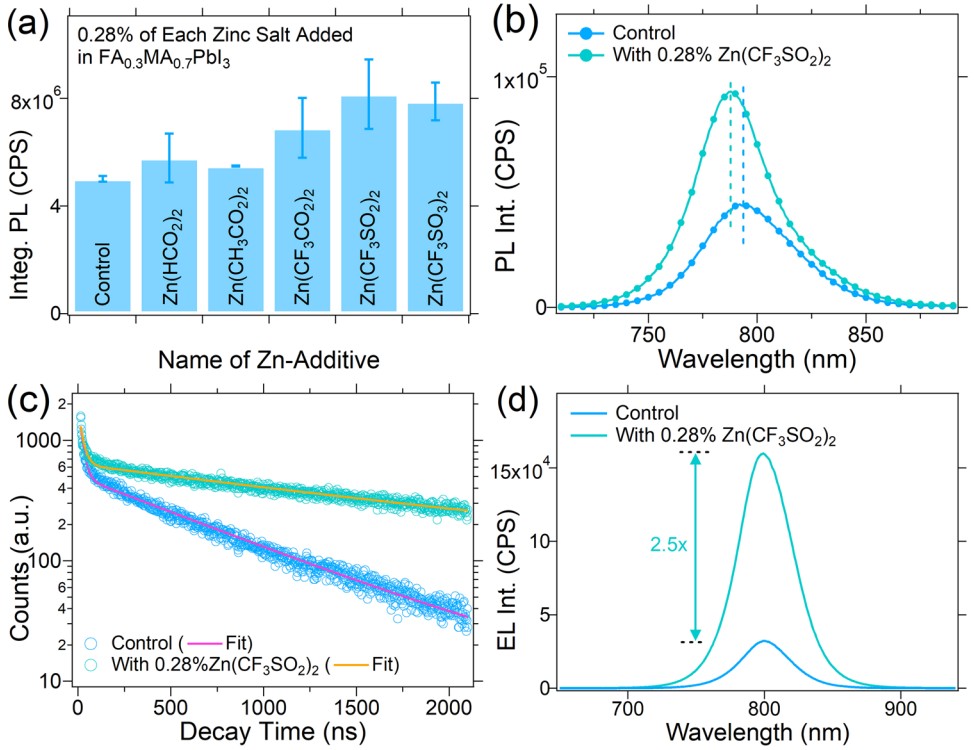

**Fig. 3 | Optical property of Perovskite thin films. a** Integrated PL intensity of control and perovskite films with different types of Zn-additives, **b** PL intensity of perovskite films without and with Zn(OOSCF$_3$)$_2$, **c** PL decay curves of perovskite films with and without Zn(OOSCF$_3$)$_2$, and **d** electroluminescence (EL) spectra perovskite devices with and without Zn(OOSCF$_3$)$_2$.

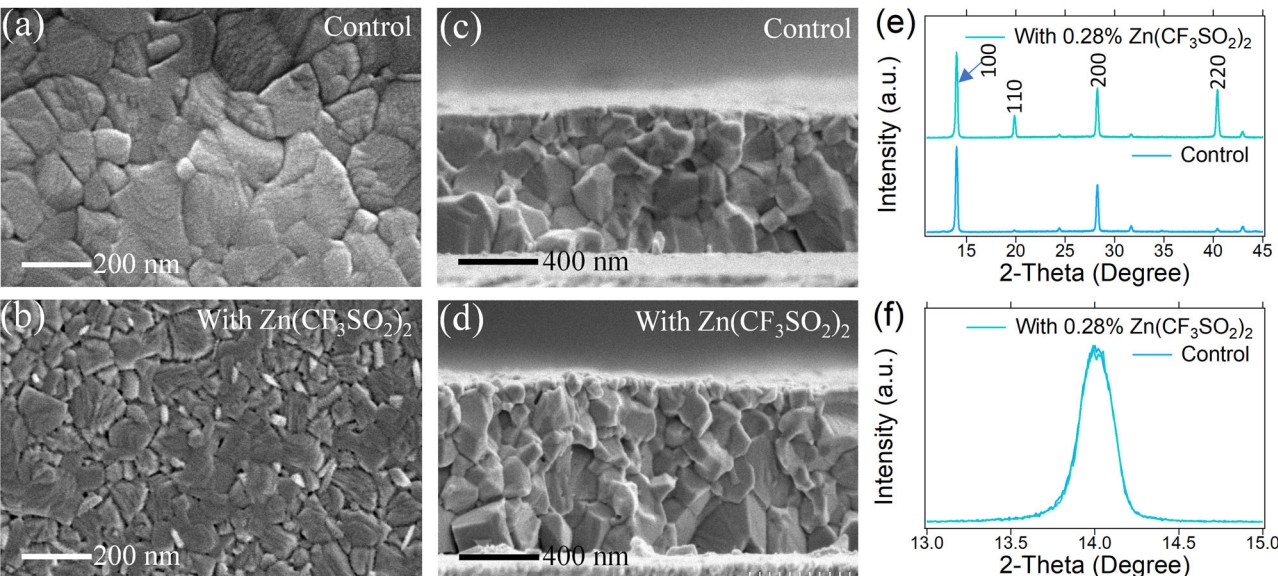

**Fig. 4 | Morphological Characterization of Perovskite Thin Films. a–d** comparison of the top surface and cross-section SEM images of control and Zn(OOSCF$_3$)$_2$ added perovskite films, **e, d** XRD patterns of control film and Zn(OOSCF$_3$)$_2$ added films showing the differences in peak intensity and broadness, respectively. **e, f** XRD patterns of control film and Zn(OOSCF$_3$)$_2$ added films showing the differences in peak intensity and broadness, respectively.

MAI and FAI from a mixture of FA$_{0.3}$MA$_{0.7}$I, we synthesized this complex by mixing FAI, MAI, and Zn(OOSCF$_3$)$_2$ in 2-ME at room temperature. The formed complex (the inset image of Fig. 5e) is rod-shaped and very crystalline, as this complex has been reported previously[38]. XRD pattern of this complex shown in Fig. 5f gives a very strong peak around 26°, corresponding to the XRD peaks in the target perovskite films and further verifying the formation of the Zn-amine complex. The Zn-amine complex may stay around the grains during the perovskite grain formations because of their very different size from ions in perovskites and passivates the perovskite grains as like how other neutral materials including PbI$_2$, MAI-MABF$_4$, and 2-D perovskites passivate[39–41].

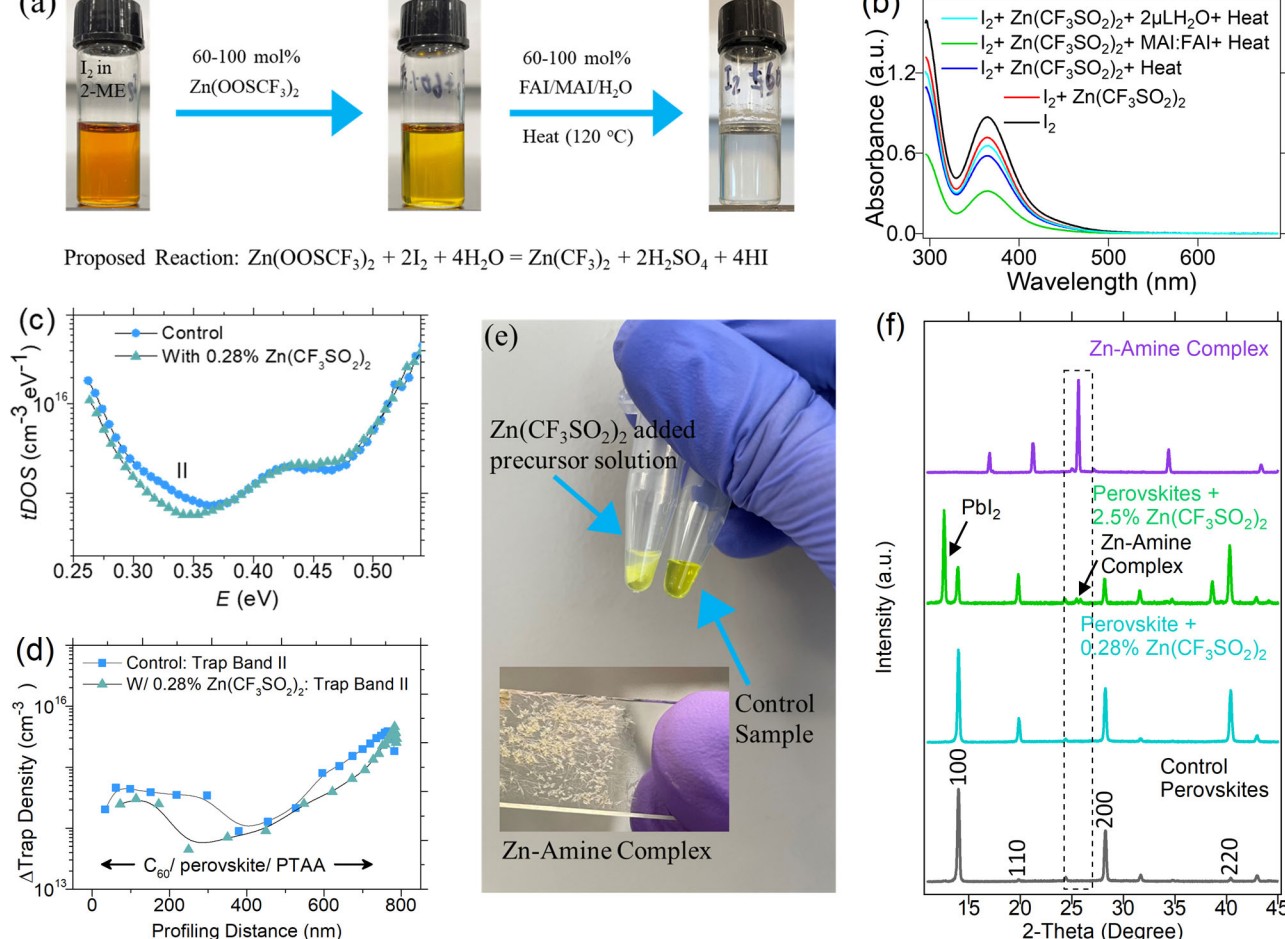

**Fig. 5 | Roles of Zn(OOSCF₃)₂ in Improving Performance. a** optical images showing the roles of CF₃SOO⁻ ions in the iodine reduction and the proposed reaction of the iodine reduction, **b** the absorption spectra collected at different conditions showing the iodine reduction of CF₃SOO⁻ ions. In the experiment, 1.9 M I₂ solution in 2-ME, 60 mol% Zn(OOSCF₃)₂ in 2-ME w.r.t the I₂ solution, and 100 mol % a mixture of FAI and MAI (3:7) w.r.t the I₂ solution were used. For the absorption measurement, respective diluted solutions were used. **c** tDOS spectra of a FA$_{0.3}$MA$_{0.7}$PbI₃ polycrystalline thin film solar cells without and with Zn(OOSCF₃)₂. **d** spatial distributions of II with the trap depth of 0.35 eV in the FA$_{0.3}$MA$_{0.7}$PbI₃ solar cells without/with the addition of Zn(OOSCF₃)₂ under a forward bias of +1.2 V measured by DLCP (note that C₆₀ or PTAA in the inset indicates the location that is close to the C₆₀ or PTAA layer of the device). **e** an optical photograph showing Zn-amine complex formation in the perovskite precursor solution when Zn(OOSCF₃)₂ is added in the perovskite precursor solution in an excess amount (e.g., 2.5 mol% to Pb). **f** XRD patterns of control film, films added with Zn(OOSCF₃)₂ at different molar concentrations with respect to the PbI₂ concentration and synthesized Zn-amine complex from a mixture of MAI, FAI, and Zn(OOSCF₃)₂.

We thus conclude that the reduction in trap density is caused by the I⁻ precipitation by zinc salts. The formation of Zn(FA$_x$MA$_{1-x}$)₄I₂ complex to take excess I⁻ ions from perovskite solution has important implications. In many perovskite solutions, iodide salts such as PEAI and DDAI are introduced to form 2D perovskites for passivation, but they inevitably introduce more iodide and thus introduce more iodide interstitials. This provides an approach to modulate the iodide concentration in perovskite solution.

In conclusion, we discovered a family of zinc additives cells that could improve perovskite solar cell efficiency and stability. Among all the zinc additives, Zn(OOSCF₃)₂ showed the best performance in passivating the defects of the perovskites despite its induction of negligible grain size changes. The CF₃SOO⁻ anions can reduce the generated iodine during perovskite solution or device aging. At the same time, the zinc cations can precipitate the excess iodide so that the iodide interstitial concentration is diminished throughout the films, resulting in improved device efficiency and stability. These additives also improve the uniformity and reproducibility of perovskite films, which facilitated the demonstration of minimodules with larger aperture areas of 80–100 cm² with a certified efficiency of 19.60%.

## Methods

### Materials

2-Methoxyethanol (2-ME, 99.8%, Sigma-Aldrich), lead(II) iodide (PbI₂, 99.99% trace metals basis, TCI), lead(II) iodide (PbI₂, 99.999%, Alfa Aesar), formamidinium iodide (FAI, 99.99%, GreatCell Solar), methylammonium iodide (MAI, 99.99%, GreatCell Solar), benzylhydrazine hydrochloride (BHC, 95%, AmBeed), 4-fluoro-phenethylammonium iodide (4-F-PEAI, >99%, GreatCell Solar), L-α-phosphatidylcholine (LP, ≥99%, Sigma-Aldrich), 5%v/v MAH₂PO₂ (MAHP, synthesized in our lab), dodecylammonium iodide (N-DDAI, GreatCell Solar), dimethyl sulfoxide (DMSO, 99.9%, Sigma-Aldrich), ZnX₂ (99.99% and ≥98%, Sigma-Aldrich), zinc formate (98%, Alfa Aesar), zinc acetate (99.99%, Sigma Aldrich), zinc trifluoroacetate (Sigma Aldrich), zinc trifluoromethane sulfinate (95%, Sigma Aldrich), zinc trifluoromethane sulfonate (98%, Sigma Aldrich), choline chloride (CC, 99%, Sigma Aldrich), poly(3,4-ethylenedioxythiophene):poly(styrene-sulfonate) (PEDOT:PSS, Clevios™ P VP AI 4083, Heraeus), poly(triarylamine) [PTAA, average M$_n$ = 7000 to 10,000, Sigma-Aldrich], toluene (TL, 99.8%, Sigma-Aldrich), C₆₀ (Nano-C Inc), bathocuproine (BCP, 96%, Sigma-Aldrich), and Cu (99.99%, Kurt J Lesker) were purchased and used as received.

### Separate perovskite solutions and additive preparation

Each 2.50 M $MAPbI_3$, 1.67 M $FAPbI_3$, 50 mg/mL BHC, 50 mg/mL p-FPEAI, 10 mg/mL LP, 5%(v/v) $MAH_2PO_2$ (MAHP), 50 mg/mL (N-DDAI), and 0.246 M CC solution were prepared separately in 2-ME at room temperature. BHC was dissolved at 110 °C through stirring for 3 hours, whereas $MAPbI_3$ and $FAPbI_3$ perovskite precursors were dissolved in 2-ME at room temperature under constant stirring overnight. 0.246 M halide Zn-additives were dissolved in 2-ME, whereas 0.173 or 0.345 M Zn-additives were dissolved in DMSO. Note that zinc formate is not entirely soluble in DMSO at room temperature when the concentration is higher than 0.173 M. So, heating the solution is necessary before using it.

### Perovskite ink preparation for blade coating

Before blade-coating, $MAPbI_3$ and $FAPbI_3$ precursor solutions, all the additives, and DMSO were mixed and diluted to a 1.4 M $MA_{0.7}FA_{0.3}PbI_3$ solution. Additives amounts were 23–25 mol% DMSO, 0.63 mol% BHC, 0.37 mol% 4-F-PEAI, 0.25 mol% choline chloride, 0.026 mol% LP, 0.000016 mol% MAHP, 0.19 mol% n-dodecylammonium iodide, and 0.14-0.55 mol% zinc salts with respect to the $Pb^{2+}$. These additives help achieve high-efficiency devices by tuning the film drying and passivating perovskites.

### Device fabrication

Fabrication of small solar cells: On the cleaned ITO, the PTAA layer (3.3 mg mL$^{-1}$ in toluene) was blade-coated with a blade gap of 150 μm above the substrate at a speed of 20 mms$^{-1}$. On the PTAA layer, the perovskite precursor ink was bladed with a blade gap of 250 μm above the ITO substrates at the same speed as the blading PTAA layer. The $N_2$ knife was set at 20 PSI during blade coating of the perovskite layer. After that, the perovskite films were annealed at 120 °C for 5.5 min in air. Solar cells were then completed by thermally evaporating $C_{60}$ (30 nm, 0.2 Å s − 1), BCP (6 nm, 0.1 Å s − 1), and 150 nm copper (1 and 2 Å s$^{-1}$) at below $1 \times 10^{-5}$ Torr.

Fabrication of submodules: For submodules, laser scribing was performed twice after Cu electrode deposition at 80 nm (P2) and 120 nm (P3) to complete the module fabrication. The fabricated modules have 20-sub cells, and each sub-cell has a width of 6.5 mm. The total scribing line width was 0.4 to 0.6 mm, giving a geometry filling factor of 92% to 94%. A polydimethylsiloxane (PDMS) layer was applied to the surface of the glass substrate as an antireflection coating. The active areas of solar cells and modules are 0.08 cm$^2$ and 78/84/108 cm$^2$. Dimension of 78, 84, and 108 cm$^2$ modules was 6.5 cm length x 12 cm width (sub-cells width: 0.65 cm and a number of sub-cells: 20 sub-cells), 7.0 cm length x 12 cm width (sub-cell width: 0.65 cm and a number of sub-cells: 20 sub-cells), 9.0 cm length x 12 cm width (sub-cell width: 0.65 cm and a number of sub-cells: 20 sub-cells), respectively.

### Film and device characterization

The current-density (J-V) characteristics of solar cells were measured using a Xenon lamp–based solar simulator named Oriel Sol3A, Class AAA Solar Simulator. The power of the simulated light was calibrated to 100 mWcm$^{-2}$ by a silicon reference cell (Newport 91150 V-KG5). All the devices were tested using a Keithley 2400 source meter with a backward scan rate of 0.1 Vs$^{-1}$ in the air at room temperature, and the delay time was 10 ms without any preconditioning before measurement. Fresh solar cells were measured within the week of fabrication. EQE spectra were obtained with a Newport QE measurement kit by focusing a monochromatic beam of light onto the devices. The photoluminescence and time-resolved photoluminescence (TRPL) lifetimes were measured using a 405 nm laser with a FluoTime 300/MicroTime 300 combined system by PicoQuant. PL spectra of the thin film samples and reference glass were collected by a PL spectrometer (Ocean Optics QEpro) equipped with an integrating sphere (Labsphere QE sphere). A continuous wave 405 nm laser was used to excite the

pure glass and samples to get PL spectra. Then $\Phi_{PL}$ was calculated following the information reported in our previous article[26]. The EL of the solar cells was determined by measuring the emitted photons of the devices in all directions through an integrated sphere by using a calibrated spectrometer (QE Pro, Ocean Optics) under a constant current density provided by a Keithley 2400 source measure unit. SEM image was taken on Hitachi S-4700 with EDS operating at 5 kV. The XRD patterns were obtained with a Rigaku SmartLab x-ray diffractometer with Cu Kα radiation (λ = 1.5418 Å) operating at 40 kV and 40 mA. The TAS and DLCP measurements were performed using an Agilent E4980A precision LCR meter. The dc bias (V) was fixed at 0 V for the TAS measurement, and the ac bias (δV) amplitude was 20 mV. The ac frequency (f) scanning range was 0.02 – 2000 kHz. The $tDOS$ ($N_T$ ($E_\omega$)) is calculated by using the equation $N_T (E_\omega) = -\frac{1}{qkT}\frac{\omega dC}{d\omega}\frac{V_{bi}}{W}$, where $q$, $k$, $T$, $\omega$, and $C$ are elementary charge, Boltzmann's constant, temperature, angular frequency, and specific capacitance, respectively. $W$ and $V_{bi}$ are the depletion width and build-in potential, respectively. The demarcation energy $E_\omega = kT\ln\left(\frac{\omega_0}{\omega}\right)$ (where $\omega_0$ is the attempt-to-escape angular frequency that equals to $2\pi\nu_0 T^2$) is derived from the temperature-dependent $C – f$ measurements as demonstrated in our previous work[35]. For the DLCP measurements, the $V$ was scanned from 0 V to the $V_{OC}$ (e.g., 1.1 V) for the perovskite solar cells. The DLCP method uses a series of variable $\delta V$ (e.g., 20 to 200 mV) to measure the junction capacitance and acquire the capacitance contribution from the trap states by taking advantage of the information embedded in the higher-order terms. The capacitance measured at each $\delta V$ was recorded and fitted with a polynomial function $C = C_0 + C_1\delta V + C_2(\delta V)^2 + ...$ to obtain $C_0$ and $C_1$. With the determination of $C_0$ and $C_1$, the total carrier density ($N$) that includes both free carrier density and trap density at the profiling distance $X$ from the junction barrier is calculated by $N = -\frac{C_0^3}{2q\varepsilon A^2 C_1}$, where $q$ is the elementary charge, $\varepsilon$ is the dielectric constant of the semiconductor which is 33 for $MA_{0.7}FA_{0.3}PbI_3$, and $A$ is the active area of the junction. The profiling distance from the junction barrier was calculated by $\varepsilon A/C_0$, which was changed by tuning the $V$. For each ac bias, an additional offset dc voltage was applied to keep the maximum forward-bias constant. The trap density within a certain trap depth range was calculated by subtracting the total carrier density measured at a larger $E_\omega$ (lower ac frequency) with that measured at a smaller $E_\omega$ (higher ac frequency). FTIR spectra were acquired with a PerkinElmer Spectrum Two FT-IR spectrometer.

### Device encapsulation and stability evaluation

PSCs were encapsulated by cover glasses sealed with epoxy encapsulant. In this process, the epoxy encapsulant is sandwiched between Cu-electrodes and cover glasses to protect all the layers while the electrodes are sticking out. A LED lamp with a light intensity equivalent to AM 1.5 G without any ultraviolet filter worked as a solar simulator in the air (relative humidity, ~50 ± 10%). The temperature of the solar cells was measured to be -50 °C due to the heating effects of the lamp. The device efficiency was measured at different times, and device temperature and relative humidity were recorded simultaneously.

We encapsulated modules with PIB edge sealing without/with commercial epoxy inside. The whole module substrate size was 15 cm 15 cm. We made the modules with target sizes of 78 cm$^2$, 84 cm$^2$, and 108 cm$^2$. So, the other space was left empty, or epoxy filled.

### Reporting Summary

Further information on research design is available in the Nature Portfolio Reporting Summary linked to this article.

## Data availability

The data that support the findings of this study are provided in the main text and the Supplementary Information. The original data are available from the corresponding author upon request.

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

## Acknowledgements

This material was mainly based upon work supported by the U.S. Department of Energy's Office of Energy Efficiency and Renewable Energy (EERE) under the award DE-EE0009529 through CubicPV Inc. Part of the module fabrication was based upon work supported by the U.S. Department of Energy's Office of Energy Efficiency and Renewable Energy (EERE) under the Solar Energy Technologies Office Award

Number 38050. The views expressed in the article do not necessarily represent the views of the DOE or the U.S. Government.

## Author contributions

M.A.U. prepared blade-coating inks. M.A.U. fabricated small solar cells. M.A.U. and P.J.S.R. fabricated large area perovskite modules. Z.N., G.Y., and M.L. carried out PL intensity, PL quantum yields, and TRPL measurements. M.A.U. and M.W. performed SEM measurements. M.A.U. and H.Z. performed XRD measurements. M.A.U. and P.J.S.R. tested solar cells and modules and studied the device stability test. P.J.S.R. and H.G. performed the PDMS film preparation and packing of modules for NREL certification. M.A.U., and J.H., wrote the paper, and B.D.D., reviewed the paper.

## Competing interests

The authors declare the following competing interest: Benjia Dak Dou is employed by CubicPV Inc., a manufacturer of solar photovoltaic equipment and materials. Huang, J. Tandem PV is an entity to which the following technologies used or evaluated in this paper have been licensed: an ink formulation for fast coating of perovskites and BHC for reducing iodine. Huang is an inventor of the technologies and has received royalties. These relationships have been disclosed to and are under management by UNC-Chapel Hill. The remaining authors declare no competing interests.
