## [Peer Review File · Nature Communications]

Iodide Manipulation Using Zinc Additives for Efficient Perovskite Solar MinimodulesEditorial Note: This manuscript has been previously reviewed at another journal that is not operating a transparent peer review scheme. This document only contains reviewer comments and rebuttal letters for versions considered at *Nature Communications*.

REVIEWER COMMENTS

Reviewer #1 (Remarks to the Author):

The authors have successfully addressed my comments in the revised version. They have clarified my inquiries clearly and incorporated them into the updated manuscript.

Reviewer #2 (Remarks to the Author):

The revised manuscript shows significantly improved quality, and I support publication of this paper as is.

Reviewer #3 (Remarks to the Author):

To control iodide-related defects in perovskite ink and films, the authors added a small amount of an organozinc chemical, zinc trifluoromethanesulfinate ($\text{Zn}(\text{OOSCF}_3)_2$). In the blade-coating process, the additives aid in the creation of more homogeneous perovskite films on large-area surfaces. Based on this technology, a certified efficiency of 80 cm² minimodules is 19.6%, which is quite promising in the perovskite community.

The authors strengthened the mechanism analysis and explained the relationship between the examined problem and the increased efficiency of solar modules in the revised version. Although the authors still lack a thorough understanding of some issues, I believe the work is worthy of publication in *Nature Communications* after the following issues are addressed.

1. In question 9 of the previous version, I asked why $\text{Zn}(\text{OOSCF}_3)_2$ works better than other organic compounds. The solution is difficult to understand. " $\text{Zn}(\text{OOSCF}_3)_2$ is highly soluble while $\text{Zn}(\text{OOSCF}_3)_2$ is sparingly soluble." Why did you chose the material if it is sparingly soluble? More specifics are required.

2. The author stated that they employed numerous more optimal passivators in the ink; what are they? The author should specify whether $\text{Zn}(\text{OOSCF}_3)_2$ inhibits crystal formation.

3. The authors agree there are many divalent cations can form metal-amine complexes, why did you chose Zn?

4. The authors showed the small devices stability, how about the stability of the minimodule?

5. The authors should update some information in the introduction, for example the championship efficiency of perovskite solar cells.

Reviewer #1 (Remarks to the Author):

The authors have successfully addressed my comments in the revised version. They have clarified my inquiries clearly and incorporated them into the updated manuscript.

Response: Thanks for the reviewer's time and effort to review and help improve this manuscript.

Reviewer #2 (Remarks to the Author):

The revised manuscript shows significantly improved quality, and I support the publication of this paper as is.

Response: Thanks for the reviewer's time and effort to review and help improve this manuscript.

Reviewer #3 (Remarks to the Author):

To control iodide-related defects in perovskite ink and films, the authors added a small amount of an organozinc chemical, zinc trifluoromethanesulfinate ($\text{Zn}(\text{OOSCF}_3)_2$). In the blade-coating process, the additives aid in creating more homogeneous perovskite films on large-area surfaces. Based on this technology, a certified efficiency of 80 cm^2 minimodules is 19.6%, which is quite promising in the perovskite community. The authors strengthened the mechanism analysis and explained the relationship between the examined problem and the increased efficiency of solar modules in the revised version. Although the authors still lack a thorough understanding of some issues, I believe the work is worthy of publication in Nature Communications after the following issues are addressed.

Response: Thanks for the reviewer's time and effort to review and help improve this manuscript.

1. In question 9 of the previous version, I asked why $\text{Zn}(\text{OOSCF}_3)_2$ works better than other organic compounds. The solution is difficult to understand. " $\text{Zn}(\text{OOSCF}_3)_2$ is highly soluble while $\text{Zn}(\text{OOSCF}_3)_2$ is sparingly soluble." Why did you choose the material if it is sparingly soluble? More specifics are required.

Response: Apologies for the typo. It would be " *$\text{Zn}(\text{OOSCF}_3)_2$ is highly soluble while zinc formate, $\text{Zn}(\text{OOCH})_2$ is sparingly soluble.*"

The description is updated as follows:

“We use 3 different functional groups, including $-\text{COO}^-$, $-\text{SOO}^-$, and $-\text{SO}_2\text{O}^-$. They all have very good binding capabilities to undercoordinated lead atoms. Among them, CF_3SOO^- and HCOO^- are more effective than others as they can reduce iodine and triiodide into iodide ions. $\text{Zn}(\text{OOSCF}_3)_2$ is highly soluble, while $\text{Zn}(\text{OOCH})_2$ is moderately soluble in DMSO. Similarly, $\text{Zn}(\text{OOSCF}_3)_2$ is moderately soluble, while $\text{Zn}(\text{OOCH})_2$ is insoluble in 2-ME. That may explain the difference among these zinc salts.”

2. The author stated that they employed numerous more optimal passivators in ink; what are they? The author should specify whether $\text{Zn}(\text{OOSCF}_3)_2$ inhibits crystal formation.

Response: $\text{Zn}(\text{OOSCF}_3)_2$ does not inhibit crystal formation, as evidenced by the XRD patterns and PL measurements. We have described all the additives and ink formulations in the SI under the subtitle perovskite ink preparation for blade coating. The section reads:

*“Before blade-coating, MAPbI_3 and FAPbI_3 precursor solutions, all the additives, and DMSO were mixed and diluted to a 1.4 M $\text{MA}_{0.7}\text{FA}_{0.3}\text{PbI}_3$ solution. Additives amounts were 23-25 mol% DMSO, 0.63 mol% BHC, 0.37 mol% 4-F-PEAI, 0.25 mol% choline chloride, 0.026 mol% LP, 0.000016 mol% MAHP, 0.19 mol% *n*-dodecylammonium iodide, and 0.14-0.55 mol% zinc salts with respect to the Pb^{2+} . These additives help achieve high-efficiency devices by tuning the film drying and passivating perovskites.”*

3. The authors agree many divalent cations can form metal-amine complexes. Why did you choose Zn?

Response: Thanks for the reviewer’s question. According to hard-soft acid and base (HSAB) theory, Pb^{2+} , Zn^{2+} , Fe^{2+} , Co^{2+} , Ni^{2+} , and Cu^{2+} ions are borderline acids. Among them, Zn^{2+} is the strongest borderline acid in the series, so Zn^{2+} was chosen to test how this strong acidic cation impacts the ink formulation and film formation.

4. The authors showed the small device stability. How about the stability of the minimodule?

Response: Thanks for the reviewer’s comments. We have tested the storage and short-light stability of the same module certified by NREL. The results of photocurrent (*J-V*) scans are shown below. So, the yellow marked text in the manuscript reads:

*“We have also tested the 6 months of storage stability of the same module certified by NREL. The module retains over 91% of its initial efficiency. We then put the same module under 1 sunlight for 24 hours, and the module retains nearly 90% of its initial efficiency. The results of photocurrent-voltage (*J-V*) scans are shown in **Fig.S6**.”*

Fig.S6. Photocurrent-voltage ($J-V$) scans show the stability of epoxy encapsulated minimodule with an aperture area of 79.67 cm^2 .

5. The authors should update some information in the introduction, for example, the championship efficiency of perovskite solar cells.

Response: Thanks for the reviewer's suggestion. We have updated the champion efficiency which is over 26%.